# Impact of Environmental and Epigenetic Changes on Mesenchymal Stem Cells during Aging

**DOI:** 10.3390/ijms24076499

**Published:** 2023-03-30

**Authors:** Nicholas Smith, Suzanna Shirazi, Dimitrios Cakouros, Stan Gronthos

**Affiliations:** 1Mesenchymal Stem Cell Laboratory, School of Biomedicine, Faculty of Health and Medical Sciences, University of Adelaide, Adelaide, SA 5001, Australia; 2Precision Medicine Theme, South Australian Health and Medical Research Institute, Adelaide, SA 5001, Australia

**Keywords:** bone marrow stromal cells, mesenchymal stem cells, epigenetics, osteogenic, bone, skeleton, DNA methylation, histone methylation, histone acetylation

## Abstract

Many crucial epigenetic changes occur during early skeletal development and throughout life due to aging, disease and are heavily influenced by an individual’s lifestyle. Epigenetics is the study of heritable changes in gene expression as the result of changes in the environment without any mutation in the underlying DNA sequence. The epigenetic profiles of cells are dynamic and mediated by different mechanisms, including histone modifications, non-coding RNA-associated gene silencing and DNA methylation. Given the underlining role of dysfunctional mesenchymal tissues in common age-related skeletal diseases such as osteoporosis and osteoarthritis, investigations into skeletal stem cells or mesenchymal stem cells (MSC) and their functional deregulation during aging has been of great interest and how this is mediated by an evolving epigenetic landscape. The present review describes the recent findings in epigenetic changes of MSCs that effect growth and cell fate determination in the context of aging, diet, exercise and bone-related diseases.

## 1. Characterisation of MSC

German pathologist Julius Cohnheim was the first to describe fibroblastoid-like cells originating from the bone marrow of rabbits and chickens with the ability to form mineralised ossicles when transplanted into muscle [1]. Almost a century later, Friedenstein and colleagues reported on the isolation and ex vivo expansion of clonal adherent stromal precursor cells from the bone marrow of rodents capable of giving rise to osteoblasts, adipocytes and chondrocytes [2]. The colony-initiating cells were denoted as colony-forming units-fibroblast (CFU-F), which were later identified in human bone marrow aspirates [2,3]. Clonal analyses of ex vivo expanded human CFU-F were also found to undergo self-renewal, support a hematopoietic stem cell niche and facilitate wound repair, as well as exhibit immunomodulatory properties [4,5,6,7,8]. These stromal precursors have subsequently been referred to as skeletal stem cells, mesenchymal stem/stromal cells (MSC), stromal precursor cells and bone marrow-derived mesenchymal stromal cells (BMSC). Although there has been major progress in identifying MSC-associated markers, no single cell surface marker has been identified as MSC-specific to date. STRO-1, CD44, CD49a, CD73, CD90, CD105, CD106, CD146, CD166 and CD271 are known to discriminate MSC from many other cell types [5,9,10,11]. Furthermore, the expression of these markers can be compromised by culture conditions and passage number [12]. Other selection markers include Leptin-receptor^+^, PDGF-receptor^+^, CXCL12^+^, α-smooth muscle actin^+^, Prx1^+^, Nestin^+^, Mx1^+^, multi-potential perivascular cells or reticular cells, with varying capacities to support haematopoiesis [5,13,14,15,16,17,18,19]. Alternate MSC isolation protocols are based on negative selection approaches by eliminating non-MSC accessory cells via the expression of antigens such as CD11b, CD14, CD19, CD31, CD34, CD45, CD45, CD79a, Glycophorin-A and HLA-DR [5,20]. More recently, CD200 [21] and Gremlin [22] have been shown to identify precursor populations in osteoblasts and chondrocytes but not adipocytes. Further investigations identified CFU-F or MSC-like populations residing in other adult tissues such as adipose, dental pulp, periodontal ligament and peripheral blood [23,24,25,26] acting as stem cell reservoirs for tissue homeostasis and repair after an injury, disease or aging. However, MSC-like populations derived from different tissues are known to have differential growth and developmental properties based on the epigenetic memory of each cell population dependent on the tissue of origin [27,28].

## 2. Biological Changes Associated with Aging MSC and the Skeleton

The skeleton is the major tissue in supporting human structure and movement. It is also considered an endocrine organ in maintaining calcium and glucose homeostasis [29]. Bone health is a major challenge in an aging populations commonly found in developed countries, where it is often linked to a more sedentary lifestyle. This has led to lower bone density in the elderly, associated with an increase in the incidence of major fractures and severe long-term disability [30,31], with mortality rates of 15–25% [32]. Therefore, aging presents as a major risk factor for skeletal disease in developed countries [33].

DNA damage, cellular stress responses and alterations in both the genome and epigenome can cause a decline in healthy tissue homeostasis and lead to a host of changes at both the cellular and tissue level. The epigenetic landscape within stem cells is a dynamic process [27,34], where the ability to undergo self-renewal and differentiation into functional, organ-specific cell types becomes increasingly limited during aging. Consequently, the functional capacity of MSC to maintain tissue homeostasis is compromised due to phenotypic alterations linked with the cellular aging and senescence of MSC, which ultimately affect their regenerative properties [35].

In postnatal organisms, BMSC retain their ability to regenerate tissues following bone fractures, articular cartilage injuries and other traumatic injuries of connective tissue. Osteoarthritis, osteoporosis and systemic inflammatory conditions such as rheumatoid arthritis are among the most common and disabling musculoskeletal degenerative disorders often associated with aging [36]. Other risk factors include obesity, alcohol intake, smoking, family history of hip fractures and joint trauma [37]. As with other stem cell populations, there are some alterations in the composition, function and structure of MSC due to both cellular intrinsic and extrinsic aging-related mechanisms [28,38]. Some intrinsic aging mechanisms that are common between all cell types include impaired DNA repair, telomere shortening, alteration in gene expression and epigenetic drift [10,28,39,40]. In regard to extrinsic mechanisms of cellular aging, changes in the tissue composition and microenvironment via hormonal and growth factor signalling, as well as an increase in inflammatory cytokines, seem to play a crucial role in MSC senescence [41,42,43]. Studies have also shown that, as MSC age, there is not only a decrease in the osteogenic and chondrogenic differentiation potential but also a shift towards adipogenic development at the expense of osteo/chondrogenic differentiation [27,41,44,45,46,47]. Therefore, bone remodelling is compromised with a shift towards bone resorption, which causes trabecular and cortical thinning with a rise in cortical porosity [48]. Furthermore, aged MSC express less pro-inflammatory factors and show weakened immunomodulatory potential, which has been associated with the early senescence of MSC [49,50]. Endogenous signals, even in the absence of any damaging stimuli, drive “age-associated low-grade inflammation known as ‘inflammaging’” [51,52], where a recent study has confirmed the role of inflammaging in declining MSC numbers and functions affecting bone healing [53]. The design of therapeutic agents to slow down MSC aging and alleviate disease is dependent on our understanding of the molecular changes that drive MSC aging. One of the pillars of aging for all organisms is epigenetic drift, where changes in the epigenome drive the deregulation in cell growth and differentiation. Discovering the key players and their modes of action have been the focus of recent studies.

## 3. Epigenetic Regulation of MSC Growth and Function

Epigenetics refers to heritable alterations in gene expression that cannot be attributed to alterations in the DNA sequence and range from DNA methylation and histone modifications to alterations in the non-coding RNA expression and chromatin structure [54]. These epigenetic alterations regulate gene expression by altering the accessibility of the DNA to transcription and accessory factors, as well as altering the expression at the translational level by interfering with mRNA (Figure 1). The result of this is a network of modifications that can shift throughout the lifespan of an organism, with a changing landscape as we interact with our environment through lifestyle choices such as diet and exercise (Table 1). These changes ultimately remodel the chromatin structure and impact stem cell biology. Interactions between lifestyle factors and epigenetic patterns in an increasingly aging population has highlighted the complex nature of “aging”, with a host of epigenetic factors being investigated into how they regulate MSC growth and function (Figure 1).

### 3.1. Structural Changes of Chromatin in Aged MSC

The relationship between the chromatin structure, nucleosome positioning and BSC aging is a recent phenomenon that is gaining traction, as its importance in transcription, DNA damage and genomic stability is essential. Recent studies have examined genome-wide chromatin accessibility in adipose-derived MSC (AMSC) using the assay for transposase-accessible chromatin with high-throughput sequencing (ATAC-seq), which found that chromatin is generally more open in AMSC compared to somatic cells [55], especially around promoters and five prime untranslated regions (5′UTR), allowing for greater plasticity in stem cells. Surprisingly, the nucleosome positions remain relatively stable during aging, as assessed by NucleoATAC-seq, indicating how robust stem cells are. Subtle changes, however, are observed around gene loci regulating stress responses. This was evident for genes encoding SUMO proteins, as sumoylation is an important post-translational modification that occurs to stress-related proteins. The nucleosome re-positioning in this region allows for a greater expression of SUMO-1 under stress, giving rise to a unique stem cell-specific protective mechanism [55]. The machinery that is central to chromatin remodelling includes the ATPase-dependent chromatin remodelling SWI/SNF family, which consists of Brahma-related gene 1 (*Brg-1*) and Brahma (*BRM*). These proteins can slide nucleosomes to either promote open or closed chromatin [56,57]. Any changes in Brg-1 expression in BMSC results in senescence, partly due to the silencing of *Nanog* expression due to the recruitment of DNA methyltransferase 1 (DNMT1) and histone deacetylase 1 (HDAC1), which methylates DNA and deacetylates histones, respectively, altering the nucleosome architecture [58]. In agreement with its role in cellular senescence, senescent BMSC have been shown to lose *Nanog* expression [59].

Another chromatin structural protein is CBX4, which is part of the polycomb repressive complex 1 (PRC1) and was found to be downregulated in aged human BMSC and in BMSC from Werner syndrome patients who exhibit premature aging [60]. Surprisingly, there was no change in genomic instability or differentiation in *CBX4* knockout BMSC, and differentiation proceeded normally; however, BMSC underwent senescence, where the RNAseq analysis identified 306 upregulated genes and 332 downregulated genes. Identification of the binding partners by mass spectrometry identified 262 proteins, most of them nucleolar proteins. CBX4 was found to bind to rDNA repeats, and its loss led to reduced heterochromatin formation in the nucleolus, together with downregulation of histone 3 lysine 9 tri-methylation (H3K9me3) and increased ribosome expression.

Gene expression profiling of old BMSC identified another chromatin structural protein, nucleosome assembly protein 1-like 2 (NAP1L2), where increased expression correlated with senescence and impaired osteogenesis [61]. NAP1L2 was found to augment chromatin accessibility by affecting histone 3 lysine 14 acetylation (H3K14Ac) by recruiting the HDAC Sirt1 to the osteogenic gene promoters and was upregulated in patients with osteoporosis. Additional studies examining chromatin structures in old BMSC using ATAC-seq identified a decrease in chromatin accessibility, with 6500 sites becoming more accessible and 8250 being less accessible [62], with the gene ontology (GO) analysis showing enrichment for cell differentiation genes. Chromatin accessibility was largely influenced by a global decrease in H3 and H4 acetylation, as well as H3K27Ac in gene enhancers. Genes that gained H3K27Ac corresponded to the metabolism. A novel mechanism was discovered where the export of acetyl-CoA from the mitochondria in the form of citrate was impaired and therefore led to reduced histone acetylation. This was evident in older and osteoporotic patients. The observed effect on aged BMSC was reversed simply by the addition of acetate, illustrating the importance of histone acetylation and chromatin structures.

More recent studies identified zinc finger protein with KRAB and SCAN domain 3 (*ZKSCAN3*) to be downregulated in aged BMSC, resulting in senescence. ZSCAN3 was found to maintain heterochromatin via its interaction with lamina proteins. Its downregulation results in the detachment of lamin-associated domains with the nuclear lamina, more accessible chromatin and the expression of repetitive sequences [63].

A recent multi-omics approach on aged BMSC, including ATAC-seq, proteomics and RNAseq, identified 2152 deregulated genes [64] with BMSC aging hotspots affecting distinct chromatin subsets. Downregulated proteins consist of the cell cycle, chromatin regulation, chromatin remodelling and RNA-processing proteins, whilst upregulated genes were involved in autophagy. Age-sensitive chromatin regions were enriched in bivalent domains, heterochromatin and repressor polycomb domains, which dictate the lineage determination of BMSC. The expression of adipogenic transcription factors increased, whilst the osteogenic factors decreased. Most importantly, aged-sensitive chromatin regions were enriched for single-nucleotide polymorphisms (SNPs) associated with bone density, body mass index and immune diseases [64], illustrating their relevance and how multiomic approaches can uncover genomic regions and proteins that drive age-associated diseases.

Although only the latest findings have been covered in this review, they reinforce previous studies that the chromatin structure is essential for BMSC self-renewal and longevity by enforcing the formation of heterochromatin, genomic stability and proper expression of differentiation genes. Chromatin remodelling and structural proteins are carefully balanced to ensure the structural integrity, and their deregulation is another layer contributing to the epigenetic drift, driving BMSC aging. Chromatin readjustments as part of epigenetic modifications that occur with aging cause changes in the gene expression profile and lineage determination, where changes in the histone methylation, acetylation, DNA methylation and hydroxymethylation are among some common epigenetic modifications between different cell types in the process of MSC aging [65].

### 3.2. Role of DNA Modifications in MSC Growth and Function

DNA modifications—in particular, DNA methylation—is a major player in the epigenetic landscape and is one of the most prolific modes of epigenetic regulation in MSC differentiation [66,67] (Figure 2). The methylation of 5-methylcytosine (5-mC) acts as the primary form of DNA modification, leading to gene silencing [66]. Cytosine-rich regions of DNA are known as CpG islands and are frequently found around promoter regions, where the methylation status of the cytosine residues determines the transcriptional activity of the gene [68]. Upon comparison between MSC derived from the various mesenchymal tissues, differences in the methylation status were found between the various cohorts based on the tissue of origin [69]. It was reported that MSC derived from adipose tissue were hypomethylated at the *PPARy* promoter whilst being hypermethylated at the *Runx2* locus. Conversely, BMSC were hypomethylated at the *Runx2* promoter, whilst *PPARy* remained hypermethylated. In addition, the methylation status at the *Sox9* locus was comparable between the two cohorts, with the methylation status only being slightly lower in the bone marrow-derived cohort.

The methylation of CpG islands in MSC plays a critical role in lineage determination, and together with the Ten-Eleven Translocase (Tet) family of DNA hydroxymethylases, controls self-renewal [65]. The three members of the Tet family (Tet1, Tet2 and Tet3) remove the 5mC mark on the DNA via the hydroxylation of 5mC to 5hmC [65] and further modifications leading to base excision repair and removal of the methyl mark. Tet1 and Tet2 have both been found to play roles in both maintaining the pluripotency in embryonic stem cells (ESC) [70] and driving the differentiation of BMSC into osteogenic and adipogenic lineages [65,71]. Tet1 has been shown to associate with NANOG to enhance the expression of pluripotency genes, *Esrrb* and *Oct4*, to maintain naïve pluripotency in ESC [72] whilst associating with Sin3a to repress the transcription of differentiation genes [73]. Conversely, Tet2 has been shown to remove silencing marks on osteogenic and adipogenic genes [65], with the knockdown of *Tet2* producing a downregulation of the osteogenic genes *Runx2*, *Bmp2*, *Osteopontin* and *Osteocalcin*. Studies on ovariectomised mice have shown decreased 5hmC levels accompanied by bone loss [71], with further studies being performed to confirm the roles of Tet1 and Tet2 in both aging and pathological bone loss.

The activity of DNA methyl transferases (DNMT) has been shown to play a role in BMSC aging, highlighting a constantly changing methylome throughout human development. Pluripotency factors such as OCT4, involved in BMSC self-renewal and growth, are impacted by DNMT activity [43]. This study showed OCT4 represses *p21* expression via the upregulation of *DNMT*, leading to increased methylation of the *p21* promoter and inhibition of cellular senescence in hair follicle mesenchymal stromal cells. Furthermore, the role of DNMTs in cellular senescence and aging has been supported, with further studies implicating decreased methylation in senescent cells [45] via the downregulation of *DNMT1* and *DNMT3b*. Other investigations reported that DNMT inhibition in MSC led to an increase in activity of cyclin-dependant kinase inhibitors, most notably *p16INK4A* and *p21CIP1*, and, ultimately, a reduction in *EZH2* and *BMI1* expression, associated with replicative senescence in MSC [74].

Recent studies have reported that the expression of the DNA N^6^-methyladenosine (N6-mA) demethylase, Alkbh1, is reduced in aging stem cells populations [34,63], with the depletion of *Alkbh1* resulting in enhanced bone loss and increased bone marrow adiposity. A significant reduction in *Osteopontin* expression, a known regulator of MSC fate determination in aging [75], was also observed in *Alkbh1* null animals. Conversely, overexpression of *Alkbh1* was reported to alleviate age-related bone loss by promoting osteogenesis. Moreover, an increase in N6-mA was observed around the promoter region of this gene, indicating Alkbh1 may regulate BMSC lineage determination through adenosine methylation of the *Osteopontin* promoter region. As well as acting as a DNA methyl mark, N6-mA also acts as an abundant mRNA modification and has been linked to cancer progression. Conversely to Alkbh1, methyl transferase-like 3 (METTL3) is a major factor in the adenosine methylation of RNA molecules and has been recently implicated in BMSC osteogenesis via post-transcriptional modification of the *Runx2* transcript [76]. In an analysis of osteoporotic samples derived from ovariectomised mice, it was found that osteoblast differentiation markers Collagen-I, Alkaline Phosphatase (ALP) and Bone Morphogenic Protein 2 (BMP2) were all downregulated compared to the control groups, as well as significantly downregulated N6-mA levels in those cohorts, along with known adenosine methyl transferases *METTL3* and *METTL4*. Upon the overexpression of *METTL3* in BMSC, it was found that osteogenic differentiation markers were upregulated, and functional differentiation assays showed an increase in ALP activity and mineral formation. The immunoprecipitation analysis showed an enrichment of a variety of miRNAs, with miR-320 being the most strongly N6-mA methylated of the cohort. Further elucidation of miR-320 targets identified *Runx2* as being a key target gene, with the overexpression of miR-320 causing a pronounced decrease in *Runx2* expression. These findings proposed METTL3 as a pro-osteogenic or anti-osteoporotic factor, with overexpression of this methyltransferase leading to the downregulation of miR-320, a now established silencing element targeting known osteogenic biomarkers such as *Runx2*.

Sclerostin (SOST) is a glycoprotein secreted by osteocytes in response to mechanical stress on the bone and has been shown to play a role in bone formation and remodelling, with the loss of SOST resulting in Van Buchem disease, associated with endosteal hyperostosis of the mandible, skull, ribs and clavicles, as well as diaphysis of the long bones [77,78]. Studies have reported an association with altered SOST expression and changes in the bone mineral density in individuals, particularly with single-nucleotide polymorphisms (SNPs) at the SOST promotor, leading to lower bone density and osteoporosis [79,80]. The regulation of established pathways such as Wnt/b-catenin by SOST [81], as well as transcription factor regulation of *SOST* by Osterix and Runx2 [79], have been well documented; however, information on the epigenetic mechanisms of gene regulation remains incomplete. It has been reported that altered methylation of the *SOST* promoter regulated the osteoblast–osteocyte transition [82], as well as a correlation between an increase in methylation and increased fracture risk in post-menopausal women. More recently, *SOST* promoter methylation was found to disrupt the transactivation of key transcription factors *Osterix*, *ERa* and *RUNX2* via the impairment of transcription factor binding [83]. The chromatin immunoprecipitation (ChIP) analysis revealed that Osterix, Era and RUNX2 bind directly to the *SOST* promoter, with a decreased binding ability found in osteoporotic patients due to elevated levels of CpG methylation.

When it comes to bone remodelling, the balance between osteoclast and osteoblast activity is critical to maintain healthy bone formation. Receptor Activator of Nuclear Factor-kappa B (RANK) and binding of its ligand (RANKL) are crucial in the development and maturation of osteoclast cells in the bone microenvironment. Conversely to this, osteoprotegrin produced by osteoblasts is able to sequester RANKL by acting as a decoy receptor in order to temper the bone resorption. The role of DNA methylation in this system has been elucidated in a study using osteoporotic patient samples [84]. The study showed an increase of the *RANKL* transcript in osteoporotic samples, coupled with a depletion of *OPG*, leading to a balance favouring bone resorption through osteoclast activity. It was found that 32 CpG sites of *RANKL* become highly demethylated in the osteoporotic group compared to the control. Accompanying this, it was found that *OPG* was largely hypomethylated in both groups, but the levels of DNA methylation in the osteoporotic group were significantly greater. These findings suggest that the imbalance of the RANKL-OPG system, caused by DNA methylation, leads to a favouring of osteoclast development and, thus, bone resorption and a bone loss phenotype.

### 3.3. MSC Aging and the Deregulation of Histone Modifications

Histone modification changes are part of the epigenetic drift that occurs in aging stem cells, which is the centrepiece of the information theory of aging [27]. This evidence came to fruition by examining whole genomes using the single-cell ChIP-seq analysis, which revealed cellular heterogeneity within the same tissue [85]. The epigenetic changes alter gene expression, regulating important signalling pathways essential for cell renewal, as well as altering the chromatin structure to compromise genetic stability, leading to transcriptional noise, where stem cells are functionally compromised and lose cell identity. Generally, histone modifications occur within the first 30 amino acids of the amino terminal domains of histones. The common residues are histones H3K4, H3K9, H3K27, H4K20 and H2AK119. The modifications include acetylation, methylation, phosphorylation, ubiquitination and sumoylation [86].

H3K27 trimethylation (H3K27me3) is a repressive mark and essential for embryonic development. As part of the bivalent domain with H3K4me3, it is crucial for the stem cell renewal and proper differentiation [87]. We and others have shown that the H3K27me3 levels increase in aged or ovariectomised murine BMSC [88,89,90,91]. The histone methyltransferase Ezh2 is responsible for H3K27me3 (45) and has been found to increase H3K27me3 on *Wnt12*, *Wnt6*, *Wnt10a*, *Runx2* and *Bglap* in ovariectomised mice, which mimics aging by altering the differentiation to favour adipogenesis at the expense of osteogenesis [89]. The enforced expression of *Ezh2* inhibited BMSC osteogenesis in an in vivo ectopic bone formation model [91], whereas the deletion of one allele of *Ezh2* in BMSC leads to increased bone volume and bone length [92]. In contrast, the deletion of both alleles leads to reduced bone volume and length, premature senescence, depletion of the MSC pool and osteoporosis in adult mice [93]. Ezh2 was found to bind and regulate the expression of novel genes *ZBTB16*, *MX1* and *FHL1* involved in osteogenesis and adipogenesis [94], as well as the cell cycle inhibitor genes, including *P15INK4b*, *P16INK4a*, *P21CIP1* and *P27 KIP1* [93].

Conditional deletion of *Ezh2* using the *Nestin* promoter as the driver results in the premature senescence of MSC, where aged MSC express reduced *Ezh2* levels compared to their younger counterparts [93]. It was further found that the downregulation of *Ezh2* results in less H3K27me3 on the promoters of *p15INK4b*, *p16Ink4a*, *p27* and *p21*, promoting senescence and decreasing self-renewal [93]. In addition, immature BMSC express higher levels of the HLH transcription factor Twist-1, which recruits Ezh2 to the *p16Ink4a* locus, repressing transcription and senescence [90]. During replicative senescence, the levels of *Twist*-*1* decrease and Ezh2 levels on the *p16Ink4A* locus decrease, leading to increased *p16* expression and senescence [90]. In another novel study, Ezh2 was shown to increase the H3K27me3 of *Foxo1*, inhibiting the antioxidant defence system [95]. By blocking Ezh2 activity, the antioxidant system was reactivated, promoting osteogenesis in aged human BMSC. The Ezh2-mediated suppression of osteogenesis was reversed when using antioxidants to eliminate reactive oxygen species (ROS) in aged BMSC.

The examination of epigenetic changes in aged human BMSC has shown that regions of the genome that acquire DNA hypermethylation are associated with repressive histone marks H3K27me3, H3K9me3 and Ezh2 enrichment [96]. DNA hypomethylated sites are associated with the active enhancer mark H3K4me1. These results show that the well-known changes in age-related DNA methylation in BMSC are regulated by specific histone marks. The H3k27me3 mark is removed by the histone demethylases KDM6A/KDM6B. the knockdown of *KDM6A* in human BMSC promotes adipogenesis and reduces osteogenesis and has been shown to bind to the transcription start site (TSS) of *Runx2* and *Bglap* [91]. The conditional deletion of *KDM6A* in early mesenchyme has shown subtle defects in bone formation in the cranium and long bones (unpublished data), suggesting redundancy with other KDM6 family members. *KDM6B* has been found to be reduced in BMSC from aged mice, and the knockout of *KDM6B* leads to severe bone defects [97]. These studies emphasised the importance of correct spatial and temporal regulations of genomic H3K27me3 in the self-renewal and differentiation of stem cells and the impact they have on mesenchymal stem cell aging (Table 2).

H3K9 methylation is also a repressive mark associated with heterochromatin, and increasing levels are associated with MSC aging [75]. An important study found that the demethylase KDM4B, which demethylates H3K9me3, H3K9me2, H3K36me3, H3K36me2, H4K20me2 and H1K26me3, is essential for BMSC self-renewal [75]. The loss of KDM4B leads to BMSC exhaustion, loss in self-renewal and promotion of adipogenesis at the expense of osteogenesis [75]. The genome-wide ChIP analysis found that KDM4B was present on *Wnt* promoters and *Runx2*, with its deletion leading to elevation of the H3K9me3 levels [75] and suppression of gene expression and osteogenesis. An additional study discovered that the H3K27me3 and H3K9me3 global levels increased in murine aged BMSC (18 month) compared to young BMSC (2 month), and the same was evident in BMSC derived from osteoporotic samples [88]. This coincided with a reduction in the H3K9me3 demethylase KDM4B and the H3K27me3 demethylase KDM6B. In another study, KDM4A was found to activate the expression of *DLX2* and *DLX3* in response to Bmp4/7, and the GO analysis confirmed KDM4B controls the genes involved in skeletal development and osteoblast differentiation. KDM6B controlled the expression of Hox genes, and the GO analysis confirmed its regulation of skeletal development and matrix organisation. A similar study found that the loss of KDM4B accelerated aging and high-fat diet-mediated bone loss due to increased senescence and BMSC exhaustion [75]. Profiling histone demethylases during the replicative senescence of BMSC identified the H3K9me3 demethylases KDM3A and KDM4C to be upregulated, whereas the H3K9me3 methyltransferase SUV39H1 was downregulated, affecting the expression of chromosome condensation genes such as condensing [98], which altered the nuclear architecture.

The transcriptional activator H3K4me3 has been shown to be downregulated in osteoporotic murine BMSC [99]. It is regulated by the methyltransferase ASH1L, a member of the Trx family, which was reported to be downregulated in ovariectomised mice [99]. In contrast, the H3K4me3 demethylase KDM5A was upregulated, leading to lower H3K4me3 levels. KDM5A was found to target *Hoxa10*, Runx2, *Osterix* and *Sox9* promoters, which displayed low levels of H3K4me3 [99]. The knockdown of ASH1L results in a decrease in H3K4me3 along the promoters of *Hoxa10*, *Runx2*, *Osterix* and *Sox9* [100].

H3K36me3 is important for transcriptional elongation and restoring the proper chromatin structure following gene transcription. The SET domain-containing 2 protein (SETD2) is responsible for H3K36me3, and its conditional deletion promotes adipogenesis and impairs osteogenesis, resulting in bone loss [101], resembling osteoporosis. Then, 983 genes show a loss of gene expression and H3K36me3 enrichment [102]. Then, 904 genes were upregulated and 913 genes downregulated [102]. H3K36me3 was found to be present on gene bodies, promoters and 3′UTR but absent from TSS.

Epigenetic rejuvenation of BMSC is becoming a new strategy to alleviate the aging phenotype. Inhibitors to HMTs and HDMs are used in the clinic in the cancer field, displaying safety and efficacy. It has been demonstrated that inhibiting Ezh2 activity using DZnep and SUV39H1 activity with chaetocin can alleviate osteoporosis. DZnep treatment can increase trabecular bone in ovariectomised mice [89], and chaetocin increases bone tissue in vivo by decreasing H3K9me3 on *Wnt* promoters [103], demonstrating that epigenetic-targeted therapies may be an effective means of rejuvenating aged stem cells.

### 3.4. Histone Acetylation Changes during Aging of Mesenchymal Stem Cells

The aging of BMSC is also associated with changes in histone acetylation (Table 2)—in particular, H3K9, K14 and K56 acetylation. The long-term culture of placental MSC results in the upregulation of HDAC4, 5 and 6, resulting in a decrease in the acetylation of H3 and H4. Acetylation of the pluripotency gene promoters *Tert*, *Oct4* and *Sox2* decreases together with the gene expression, resulting in a decrease in stem cell self-renewal [104]. Studies in adipose-derived MSC show that, during replicative senescence, the levels of HDAC1 and 2 decrease together with the polycomb group proteins (PcG). This promotes H3 acetylation of the *KDM6B* promoter, which was verified by the knockdown of *HDAC1* and *2*. The increase in KDM6B results in a decrease in H3K27me3 along the *p16Ink4A* locus increasing expression and senescence [105]. During BMSC senescence, it was found that the expression of HDAC1 and 2 decreases rapidly. HDAC1 and HDAC2 upregulate the expression of *Ezh2*, *BMI1* and *SUZ12* and downregulate *KDM6B* [105]. The downregulation of HDAC1 and HDAC2 decreases the expression of *Ezh2*, *BMI1* and *SUZ12* and increases *KDM6B*, reducing the H3K27me3 levels on the *p16Ink4A* promoter, promoting senescence.

H3K9 acetylation has been shown to be drastically downregulated in aged BMSC [106] and is deposited by general control nonderepressible 5 (*GCN5*), p300/CBP-associated factor (*PCAF*) and removed by HDAC9. H3K9 acetylation is a primary target for GCN5, which has been shown to reduce dramatically in aged BMSC, and in osteoporotic BMSC from ovariectomised mice [107]. Furthermore, the siRNA-mediated knockdown of GCN5 compromises H3K9 acetylation, especially on the promoters of Wnt genes, compromising osteogenesis [107]. Although PCAF has been shown to acetylate H3K9 on BMP2, *BMP4* and *Runx2* promoters [108], its role in aging is yet to be determined. A similar study showed the expression of HDCAC9 to increase during age-related bone loss coinciding with a decrease in H3K9 acetylation. This resulted in impaired osteogenesis, which was associated with abrogated autophagy in BMSC. The gene expression levels of key autophagy genes were downregulated in aged BMSC, with the increased binding of HDAC9 to autophagy genes *ATG7, BECN1*, *Lc3a* and *Lc3b* in aged BMSC [106]. This shows the importance of autophagy and the regulation of autophagy by histone acetylation in driving osteogenesis. Pluripotency genes also show dramatic changes in acetylation during BMSC aging, with a decrease in H3K9 and K14 acetylation seen on promoters for *Oct4*, *Sox2* and Tert, resulting in decreased expression [109] and compromising their self-renewal.

Another important acetylation mark is H3K56 acetylation, which increases in aged BMSC. One of the key enzymes essential for depositing this mark is the NAD+-dependent Sirt6 enzyme [110]. The deletion of Sirt6 in BMSC results in the reduced expression of the antioxidant *HO-1*, as SIRT6 is responsible for H3K56 deacetylation on this promoter; an increase in ROS, oxidative stress and impaired osteogenesis and chondrogenesis. Senescence is accelerated due to the increased expression of *p16* and *p21* [111].

Since senescence is a hallmark of aging, and the ablation of senescent cells in a whole animal prolongs the overall lifespan [74,112]. A recent study performed a genome-wide CRISP-Cas9 loss of function screen using precursor mesenchymal cells from a premature aging mouse model known as Werner syndrome [113]. Approximately 19,000 coding genes were targeted, where long-lived cells following serial passage indicated an abrogation of senescence. The cells that escaped senescence displayed youthful features, including increased proliferation, reduced senescence and restored nuclear architecture. Knockdown of the histone acetyl transferase *KAT7* reduced senescence and restored the ability of the passaged cells to undergo osteogenic differentiation. Heterochromatin formation was also restored, resembling young BMSC. One-third of the deregulated genes were restored following *KAT7* knockdown, including the genes involved in chromosome organisation and DNA packaging. In contrast, the overexpression of *KAT7* induced heterochromatin loss, increased *p16* and *p21* expression and was found to acetylate H3K14 of the *p15Ink4b* promoter. Moreover, *KAT7* and *p15Ink4b* were found to be increased in aged human BMSC (79–81 years old) versus young BMSC (16–28 years old). Interestingly, lentiviral-mediated deletion of *KAT7* in aged mice targeting the liver increased the overall health and lifespan of these animals. Similar screens have identified the acetyltransferase p300 [114] to prolong the lifespan, whereas the knockout of *SIRT6* leads to an increase in H3K9Ac and H3K56Ac in association with increased rates of aging [111,115].

Following transcription, the closed chromatin state is restored by various complexes, including the histone demethylase Kdm5b [116,117] and the DNA methyltransferase Dnmt3b [116]. These proteins are recruited by H3K36me3, and any deregulation in this process leads to cryptic transcription, where transcripts are made from internal TSS. An interesting study used culture-expanded MSC as an in vitro model of aging to examine exon-based transcription, where downstream exons displaying higher transcripts per kilobase million than the first exon indicates cryptic transcription. Highly expressed genes were found to have cryptic transcription in aged MSC and HSC [118]. DAVID clustering revealed clusters related to telomere maintenance, protein folding and translation and endoplasmic reticulum function, indicating that cryptic transcripts will interfere with these functions. Using the ChromHMM analysis, the chromatin structure was examined to discover the greatest change was a loss of constitutive heterochromatin; gain of facultative heterochromatin and decrease in overlap between H3K4me1, H3K27Ac and H3K36me3, while H3K4me1 and H3K27Ac distribution is wider. H3K36me3 in gene bodies becomes depleted. Interestingly, H3K36me3 restores the chromatin structure once RNA polymerase II moves through. Its depletion will lead to a more open conformation, promoting cryptic transcription. Further analysis discovered that the TSS around cryptic start sites also contained H3K4me1, K27Ac and K4me3 and less CpG DNA methylation, allowing a more permissive chromatin state. Given that the cryptic transcripts are involved in telomere maintenance and proteostasis, it is predicted that these systems are compromised during BMSC aging.

### 3.5. Non-Coding RNAs (miRNA, lncRNA) Regulation of MSC

Non-coding RNAs, such as miRNAs and long non-coding RNA (lncRNA), are functional RNA molecules that are not translated into proteins but are able to regulate the expression of genes post-transcription [119]. These non-coding RNAs are transcribed, before modification by enzymes DROSHA and DGCR8 in the nucleus, then further modified in the cytoplasm by DICER, allowing for mature miRNA to silence and degrade functional RNAs and silence the protein expression of target genes [119].

Various miRNAs are able to exhibit control over MSC differentiation during a number of stages in the process through regulation of the master regulators (Runx2, Sox9 and PPARγ) [120,121,122] via osteogenic markers such as ALP [123] and Osx [124] in developing osteoblasts. Targeting of the master regulators has been well documented, with three master regulators listed above all able to be targeted by miRNA. Families of miRNA are known to potently target Runx2 [125], such as miR-23a and miR-30c, to control early osteogenesis, while miR105 has been shown to inhibit *Sox9* to enhance osteogenic differentiation over chondrogenesis [121]. Therefore, it appears that epigenetic control over these transcription factors in both the early and late stages of bone cell differentiation is an essential process in regulating osteogenesis, which additionally requires coordinated regulation by miRNA targeting different signalling components mediating BMSC cell fate determination.

Investigations into miRNA since their discovery [126] have been categorised through the depletion of DGCR8 in ESC before rescue via miRNA mimics [127] to elucidate the roles of a host of miRNAs in the cell cycle and pluripotency, amongst others. Using this technique, studies have shown how miRNAs miR-29a-3p and miR-30c-5p can affect MSC senescence and cellular proliferation through known epigenetic regulators SOD2 and DNMT3A [128]. DGCR8 and, as such, global miRNA depletion in human MSC caused a significant reduction in cell proliferation, along with enhanced senescence and increased ROS production through mitochondrial dysfunction compared to untreated cells. Comparisons of DGCR8-depleted cells and cells that have undergone replicative senescence showed the same canonical senescence pathways shared between the two cohorts, supported by a bioinformatics analysis, supporting the induction of senescence via miRNA depletion. *SOD2* was found to be dysregulated upon *DGCR8* knockdown, playing a role in enhanced ROS production and premature senescence. Elucidation of the mechanism discovered hypermethylation at the SOD2 promoter after DGCR8 loss, and upon further investigation, it was found that DNMT3A was upregulated in these cells, suggesting that DNMT3A is the intermediate regulator of SOD2 expression. Using predicted conserved miRNA families, it was found that miR-29 and miR-30 share high sequence homology, and upon exogenous expression in DGCR8 KO cells, were able to reduce the ROS, as well as decrease the senescence factors and increase the cell proliferation rate. Through the regulatory control of BMSC proliferation and ROS production, miR-29a-3p and miR-30c-5p can elicit control over the early stages in differentiation.

A number of miRNAs have been directly implicated in regulating osteogenesis, targeting many different genes and signalling pathways. Investigations have shown the relationship between different miRNAs and the activity of TWIST-1, a key transcriptional regulator of BMSC self-renewal and growth, as a suppressor of osteo/chondrogenic differentiation [123,129]. TWIST-1 has been shown to positively regulate miRNAs miR-199a and miR-124, inhibiting osteoblast function via Wnt signalling (miR-199) and the FGFR1, JNK and p38 pathways (miR-124) [130,131]. In addition, TWIST-1 was also found to upregulate miR-376c-3p in BMSC and cranial bone cells during growth and osteogenic differentiation [123]. It was reported that the inhibition of miR-376c-3p induced an increase in BMSC and cranial cell osteogenic potential, whilst the overexpression of miR-376c-3p suppressed cell proliferation and differentiation via the regulation of IGFR1/Akt signalling to inhibit the expression of key osteogenic genes such as *Runx2*, *ALP*, *OPN* and *OCN*. This presents miR-376c-3p as a regulator of mature osteoblast function and differentiation from the undifferentiated MSC form all the way until the mature osteoblast stage.

Other miRNAs have also been shown to play direct roles in bone development and during osteoporosis. Studies have described miRNA-19a-3p as playing a role in osteogenesis during normal and pathological conditions [132]. Prior to this, the clusters of miRNAs (miR-17-92), which include miRNA-19a-3p, have been reported to play a role in a number of malignancies, such as breast [133], lung [134] and gastric [135] cancers. Through transcriptome analysis, miR-19a-3p was shown to be downregulated in osteoporotic patient samples, whereas miR-19a-3p was upregulated in human MSC cultured under osteogenic inductive conditions. The predicted binding sites uncovered HDAC4 as a putative miR-19a-3p, part of the histone deacetylase family tasked with removing acetyl marks from histones and restricting gene expression via chromatin remodelling, adding a layer of control at the mature osteoblast stage [136]. The overexpression of miR-19a-3p was found to negatively regulate the protein level of HDAC4, leading to the increased expression of osteogenic markers such as *ALP* and overall osteogenesis through chromatin remodelling. A further study implicated the role of miR-181a/b-1 in osteogenesis and bone development, albeit through a different mechanism of action [137]. The data demonstrated an upregulation of miR-181a/b-1 in hypertrophic chondrocytes in long bones, suggesting a role in endochondral bone formation and chondroblast differentiation. An analysis of the cells taken from miR-181a/b-1 null mice identified the phosphatase and tensin homolog (PTEN) as a potential target gene, a known inhibitor of the PI3/AKT signalling pathway. *PTEN* knockout mice display an enhanced bone formation phenotype, where PI3K/Akt signalling is central in osteoblast differentiation and, thus, bone development [138,139]. Another key player in osteoblast differentiation and osteogenesis is Osx, a transcription factor essential for bone development [140]. Similar to the other factors, Osx can be modulated via epigenetic mechanisms, with miR-138 being a key established miRNA involved in the regulation of its activity via the Runx2 pathway [141] at the MSC to pre-osteoblast stage. The effect of miRNA regulation on a wide array of different factors spanning the lifespan of osteoblasts from MSC to functional osteoblasts illustrates the volume of regulation factors that play a role in bone development (Table 3).

Comparatively to miRNA, long non-coding RNA (lncRNA) are longer transcripts of more than 200 nucleotides associated with gene regulation in many cellular processes, such as proliferation, differentiation and cancer development [142,143]. The function of this family of RNAs is not yet fully understood; however, they have been shown to regulate gene expression via assistance in transcription factor-promotor binding [144]. A novel lncRNA was identified that is able to regulate the fate of BMSC during aging, denoted *Bmncr* (bone marrow stem cell-related lncRNA) [112], promoting bone formation. *Bmncr* was found to be highly expressed in bone and adipose tissue, displaying a high expression in the BMSC of young mice, with a decrease in older mice. The deficiency of *Bmncr* produced a premature bone aging phenotype in a KO model, with the overexpression leading to increased bone density and reduced bone marrow fat. It was revealed that *Bmncr* upregulates the extracellular matrix molecular fibromodulin to anchor BMSC in trabecular-rich regions via activation of the BMP2 pathway. Furthermore, it was elucidated that *Bmncr* interacts with TAZ and ABL, facilitating the TAZ and RUNX2/PPARy transcriptional complex, further promoting osteogenesis whilst inhibiting adipogenesis in bone. These findings demonstrate that non-coding RNA species play important roles in bone development and have helped identify potential targets for the rescue of osteoporosis and other bone phenotypes, with a multitude of pathways and factors involved in these complex systems.

**Table 1 ijms-24-06499-t001:** The key epigenetic modifications in MSC and their modes of gene regulation.

Epigenetic Modification	Mode of Gene Regulation	Reference
DNA Methylation	DNA Modification (Cytosine/Adenine methylation)	[66]
Histone Methylation	Histone modification	[85]
Histone Acetylation	Histone modification	[85]
miRNA	Targeting of mRNA	[118]
Long non-coding RNA	Targeting transcription machinery	[142]
Chromatin Structure	Altered gene accessibility	[54]

**Table 2 ijms-24-06499-t002:** Summary of the key histone modifications related to function and abundance in aging MSC.

Modification (Function)	Writers/Erasers	Effect of Aging on Modification	Reference
Methylation			
H3K27me3	Ezh2	Decreased	[87]
H3K9me3	SUV39H1	Increased	[74]
	KDM-3A/4C		[97]
	KDM4B		[100]
H3K36me3	SETD2	Decreased	[98]
H3K4me3	ASH1L	Decreased	
	KDM5a		
Acetylation			
H3K9Ac	GCN5	Decreased	[105]
	HDAC9		[106]
H3K56Ac	SIRT6	Increased	[109,110]
H3K14Ac	KAT7	Increased	[61,112]

**Table 3 ijms-24-06499-t003:** Summary of miRNA implicated in MSC and skeletal tissue aging and development.

miRNA	Function and Pathways (Reference)		
miR-29a-3p	Maintenance of stem cell self-renewal	SOD2 and DNMT3A	[127]
miR-30c-5p		Inhibition	[127]
miR-19a-3p	Induction of Osteogenesis	HDAC4 Inhibition	[131]
miR-181a/b-1		PI3/Akt	[136]
miR-105		Sox9	[120]
miR-199	Osteogenic Inhibition	Wnt Signalling	[126]
miR-124		FGFR1, JNK, p38	[130]
miR376c-3p		IGFR1/Akt	[122]
miR-23a		Runx2	[124]
miR-30c		Runx2	[124]
miR-138		Runx2/Osx	[140]

## 4. The Role of Environmental Factors in MSC Aging and Disease

Type II diabetes mellitus (T2DM) is among the most common global chronic diseases, with a prevalence of 8.5% in adults [145]. The global rise in T2DM prevalence is allied with the obesity pandemic, which, in turn, is coupled with the consumptions of high-fat diets (HFD) [146]. The development of insulin insensitivity in people with obesity is associated with a higher prevalence of fragile bones due to osteoporosis and increased risk of major fractures [147]. Diets that are particularly high in saturated fat can negatively influence the bone mineralisation and skeletal integrity in both human and animal studies [148,149,150]. Interestingly, the fat quality seems to be as important as fat quantity in changing skeleton characteristics, where diets containing higher saturated fats have more adverse effects on bone development compared to those diets containing high levels of unsaturated fats [148,151].

Maternal obesity in mice caused by high-glucose/fat diets has been shown to compromise the skeletal health of dams and during foetal development and postnatal growth in offspring, increasing the risk of low bone mass in adulthood [152,153,154]. Furthermore, investigations of foetal MSC derived from the offspring of obese mothers in human and rodent studies showed increased cellular senescence signalling, with the overexpression of *p53* as a hallmark of senescence associated with a significant reduction in glucose metabolism [155]. These findings implicate the role of the diet in altering MSC growth and properties via diet-mediated epigenetic changes during pregnancy.

### 4.1. Effect of Diet on Epigenetics in MSC

During prenatal and postnatal development, epigenetic modifications have a crucial role in regulating the fate of different cells. Diet controls the epigenetic regulations of gene expression. Offspring born to HFD-fed mothers have altered epigenetic regulation in bone development and formation. Various studies have shown the effect of maternal nutrition on the metabolism status of offspring, which mainly occurs through epigenetic alterations and the gene expression of key metabolic pathways, including glucose metabolism [71,87,156,157].

H3K27me3 has been shown to be a repressive epigenetic histone modification of RUNX2, the master transcriptional regulator of BMSC osteogenic differentiation [158]. Maternal obesity has been associated with an increase in the expression of senescence genes of osteoprogenitors through the acetylation of H3K27ac [159]. The expression of osteoblastogenesis-associated genes such as *Pthlh* and *Col2α1* was synergistically decreased through significantly enriched H3K27me3 and H3K27ac in dams of HFD-fed maternal mice [160]. The diet has also been shown to influence the nutrition status of the cells, which provides different metabolites for epigenetic activities. For example, acetyl-coenzyme A, S-adenosylmethionine and nicotinamide adenine dinucleotide (NAD) are among many metabolites that are essential cofactors in histone methylation and/or acetylation affecting gene expression [161].

Diet and, more specifically, glucose implement their effects on bone formation through “the bone–pancreas loop”. Osteocalcin (OCN) is a major bone-specific protein secreted by osteoblasts in a carboxylated form. The acidic environment created by the maturation and activity of osteoclasts promotes the decarboxylation and activation of OCN and its release into the circulation. Upon binding to its receptors on pancreatic β cells, OCN triggers insulin release into the circulation. It stimulates β-cell proliferation and increases insulin sensitivity in liver, muscle and adipose tissue, in addition to its role as a metabolic hormone. Insulin binds to its receptors on osteoblasts (InsR) and increases the glucose uptake [162,163,164]. Insulin mainly controls OCN production by osteoblasts through two mechanisms. Firstly, by blocking the activity of Twist-2, an inhibitor of RUNX2, and driving OCN production. Secondly, by negatively controlling the activity of osteoclasts to increase the resorption of the bone matrix by creating a low pH environment [165,166]. Therefore, an endocrine feedforward loop is created between the bone and the pancreas, where insulin controls the activity of OCN while osteoblasts regulate insulin secretion from the pancreas via the release of OCN [167]. Thus, the negative correlation between the serum levels of uncarboxylated OCN and plasma glucose concentrations can be used as a measure of the osteoblast activity [168,169,170].

Studies of *Ocn-*deficient mice showed glucose intolerance, hyperglycaemia and diabetes-like characteristics in these mice [163]. Thus, it can be hypothesised that high blood glucose is potentially a cause of bone loss and bone disease. Various investigations have reported on the effects of HFD and T2DM on the quality of the bone and the development of osteoporosis [4,28,38,39,83]. Studies of BMSC derived from HFD-induced obesity and T2DM individuals reported a reduction in BMSC proliferation and osteogenic potential associated with increased apoptosis and low bone quality [4,41]. Moreover, a partial or fully inducible ablation of osteoblasts in mice led to insulin insensitivity and an increase in the blood glucose levels [42]. In addition, high glucose levels have been shown to direct BMSC towards an adipogenic fate rather than undergoing osteoblastic differentiation. While insulin insensitivity drives the reduction of osteoblasts numbers and bone formation in hyperglycaemic patients; the accumulation of lipid-dense adipocytes in the bone marrow of long bones also contributes to thinning of the cortical envelope, causing more fragile bones, akin to an osteoporotic phenotype [171]. Although high blood glucose levels are linked to a high fracture risk in metabolic patients such as diabetes, the epigenetic changes that occur in the skeletal tissue of these patients remain largely unknown.

A high protein intake is directly associated with a high bone density. The positive impact of a high protein intake on bones occurs through several mechanisms such as calcium absorption and IGF-1 secretion. However, the epigenetic role of a high-protein diet and HFD on the fate of MSC has yet to be fully investigated. Studies on both human and animal BMSC have demonstrated the essential role of protein intake on bone health during aging through regulation of the calcium balance and metabolism in the presence of both high and low-protein intakes [172]. Essential amino acids (AA) availability determines BMSC patterns of osteoblastic proliferation and differentiation. BMSC grown in a decreased AA concentration showed early senescence and cell death [173]. Whilst AA availability in low-protein intake (LPD) has been shown to impair bone mass during growth through IGF-I with age [174], there is little known about how LPD effects the overall bone health and BMSC fate differentiation in young adults [175]. Other studies of mouse maternal protein intakes have also shown to affect foetal bone formation and mineral quality; yet, these findings were gender-specific and inconsistent among all experimental groups [176]. Furthermore, it was reported that LPD caused an increased expression of DNA methyltransferases Dnmt1 and Dnmt3L [177], which may potentially act as a driver for regulating bone formation.

### 4.2. Effect of Diet on MSC during Aging

Caloric intake has a significant impact on stem cell self-renewal and lineage differentiation, as well as aging. While caloric restriction (CR) has been demonstrated to increase the lifespan, stemness and delay aging, HFD has been associated with the acceleration of aging in stem cell populations. In addition, the formation of ROS and DNA damage, as the result of HFD and the metabolic status, is also a major contributor to the acceleration of cellular senescence and the aging process. In contrast to HFD, CR has been reported to enhance stem cell function and decelerate cellular senescence in mouse HSC [178].

Obesity and aging are considered important intrinsic and extrinsic stressors that apply epigenetic effects on the gene expression of stem cells and alter their fate. Adult stem cells are capable of acquiring epigenetic memory from these stimuli over time. Aging is uniquely linked to the hypermethylation of CpG islands and hypomethylation of global DNA in all cell types [179,180]. One study identified the role of either aging or obesity in global DNA hypomethylation and the synergic effect of these two factors on lean and obese mice. The results confirmed that the global hypomethylation of adipose-derived stem cells occurs with aging, and the transcriptome remains extremely stable in lean mice, while obesity impairs this process [61]. Moreover, alterations in the cellular metabolic state as the result of aging claim to influence an organism’s stem cell lifespan and affect the activation of epigenetic enzymes with their availability.

The metabolic state plays a vital role in MSC function and survival. Age-associated changes in the MSC metabolic clock occur via mitochondrial dysfunction and accelerated anabolic signalling, causing an increase in the production and accumulation of ROS and cellular senescence [181,182]. In addition, recent reports have shown the effects of cellular metabolism on epigenetic states and organism aging [183,184]. It was observed that NAD^+^ levels, a cofactor of Sirtuins (SIRT1–7) deacetylases, decreases with age, causing a reduction in cellular histone acetylation [185,186]. As MSCs reside within a hypoxic environment, glycolysis is the major source of energy production, with a low ROS production at a steady rate [187,188]. Proliferation and aging shift the MSC metabolism towards oxidative phosphorylation, which drives the cellular senescence by increasing the ROS level and oxidative stress [189]. Higher levels of ROS damage the protein and DNA, especially at telomeric regions. This is in parallel with the decrease in SIRT1, which, in turn, impedes FOXO1, a transcription factor regulating antioxidant expression, deacetylation and repress antioxidant machinery [183,190].

NAD is also a cofactor of histone modifiers, including H3K4me3, H3K36me3 and H3K27me3, influencing epigenomic regulation [105,183]. Furthermore, BMSC of aged donors have been shown to have a lower expression of *Hox* genes as a regulator of DNA methylation. Consequently, DNA was hypomethylated, changing the differentiation and regeneration potential of these cells [27,191].

### 4.3. Mechanical Loading Effects on MSC

Exercise or mechanical loading is a key factor in improving the general body health through balancing metabolism and cardiovascular functions, as well as promoting good skeletal health. Physical activity can prevent bone deterioration by antagonising RANKL signalling, leading to decreased bone resorption by OC and increasing bone formation and strength with reduced adiposity in the marrow cavity [192,193,194]. A number of studies have also reported how the mechanical stimulation of BMSC leads to reductions in adipose formation and upregulation of osteogenesis via increased levels of osteoblastic markers such as RUNX2 [164,195,196,197,198,199,200,201]. Conversely, bone formation was reduced and osteoclast activity was enhanced during inactivity [202].

The effect of mechanical force on BMSC gene expression profiles through epigenetic modifications has been explored in recent years. Actin polymerisation has been shown to regulate epigenetic gene regulation by changing the physical shape of nuclei, which affects specific signal transduction [203]. Moreover, mechanical force appears to drive BMSC differentiation towards the osteoblastic lineage, in part through the induction of non-coding RNA, such as lncRNA-MEG3 [104]. According to another similar study, miRNA-103A and its host gene *PANK3* are downregulated following mechanical stimulation in BMSC, which stimulates the expression of the *Runx2* protein [204]. In addition, the result of a study on human BMSC showed the elevation of bone growth under mechanical stress due to the reduction of DNA methylation of osteogenic genes, causing an increase in expression [205]. Furthermore, the analysis of human skeletal muscle showed alterations in methylation at 4919 sites of DNA following exercise [206]. Similar reports described that DNA methylation was altered in a genome-wide manner in human fat tissue following exercise, linking exercise with the epigenetic regulation of adipocyte metabolism [207]. Similar reports described that DNA methylation was altered in a genome-wide manner in human fat tissue following exercise, linking exercise with the epigenetic regulation of adipocyte metabolism. One study found that, following exercise, 17,975 individual CpG sites showed differential DNA methylation in adipose tissue. Several of these genes have been previously associated with obesity and T2D [207].

The effect of a running exercise was further investigated in osteoporotic patients and in an ovariectomised mouse model of osteoporosis. The findings identified nuclear factor erythroid-derived 2-related factor-2 (Nrf2) as a critical anti-osteoporotic factor that was heavily suppressed by the activity of DNMT in its promoter. This study claimed that a running exercise could correct the effect of DNMT hypermethylation of the *Nrf2* promotor, making this epigenetic feature a potential target of the pathogenesis of osteoporosis [204].

It has also been proposed that exercise can control bone formation through epigenetics by regulating inflammation and triggering changes in the gene expression patterns of neutrophils and peripheral blood mononuclear cells, which can lead to the inhibition of bone resorption [208]. Another study examined the effect of physical activity on the methylation of human apoptosis-associated speck-like protein (ASC), a p53-target gene that regulates the p53-Bax mitochondrial apoptotic pathway and is linked to IL-1β and IL-18 secretion [209]. IL-1β and IL-18 are proinflammatory cytokines that have a prominent role in the development and progression of several age- and inactivity-related diseases. The role of aging and exercise on CpG island methylation of the *ASC* gene revealed that, although the expression on the *ASC* gene is upregulated by age, six months of intermittent walking reduced *ASC* expression by increasing DNA methylation, which consequently lowered the level of proinflammatory cytokines [209]. Therefore, it appears that environmental and endogenous cues alter the gene expression pattern and, ultimately, the fate of BMSC through epigenetic regulation. Considering the reversible nature of epigenetic changes, modification of the MSC epigenetic landscape via physical training or external stimuli may be an appropriate target for reprograming MSC gene expression to improve cell survival and function.

## 5. Conclusions

The information theory of aging relates to the erosion of the epigenetic landscape as cells age. This contributes to the deregulated function of cells and their loss of identity. Investigation of the epigenetic changes in MSC as they age over the years has demonstrated dynamic changes in DNA methylation, hydroxymethylation, histone modifications and non-coding RNA. These changes result in senescence, skewed differentiation favouring adipogenesis at the expense of osteogenesis and the loss of heterochromatin. The chromatin structure is now recognised as an important guardian of the genome and a transducer of epigenetic information that is altered during aging, where MSCs have a higher level of accessible chromatin compared to somatic cells to ensure plasticity. Interestingly, MSCs show remarkable nucleosome position stability as they age, with changes occurring along stress response genes to ensure protective mechanisms are in place. Changes in the chromatin structure and suppression of stemness genes through decreased chromatin accessibility, especially along differentiation genes enriched for bivalent chromatin domains, drive MSC aging. The advantage of assessing chromatin accessibility is amplified with the discovery that chromatin changes seem to occur in regions with identified SNPs for multiple diseases, including obesity, immunity and bone-related disorders, emphasising another level of exploration for potential therapeutic targets to counter stem cell aging and disease.

Histone and DNA modifications are the key mediators of the chromatin structure and genomic regions associated with MSC age-related increases in DNA methylation in association with repressive histone marks H3K27me3 and H3K9me3, which also appear to play important roles during osteoporosis. Of note, Ezh2 has been shown to deposit H3K27me3 on bone-associated genes in MSC derived from osteoporotic samples, as well as deactivate the antioxidant defence system, reducing osteogenesis. Interestingly, supplementing antioxidants or inhibiting Ezh2 can reverse the effects on aging MSC and restore differentiation. Moreover, the H3K9me3 levels also increase during MSC aging, and loss of the H3K9me3 demethylase *KDM4B* results in repression of the osteogenic-related genes, increased MSC senescence and accelerated aging. Many studies have confirmed a global decrease in H3 and H4 acetylation mainly due to an increase in *HDAC4*, *5* and *6*, resulting in a decrease in stemness gene expression and self-renewal. In particular, H3K9 acetylation decreases during aging and is associated with the decreased expression of *Wnt*, *Bmp* genes and *Runx2*, compromising MSC osteogenesis and autophagy. HDAC1 and 2, on the other hand, increase in expression, leading to increased *p16* expression and MSC senescence. These examples illustrate how the erosion of the epigenetic landscape alters transcription at multiple levels in aging MSC.

The additional layer of epigenetic modifiers in the form of miRNA and long non-coding RNA has also advanced considerably in recent times, as they have been shown to be deregulated in aged MSC. As highlighted in this review, miRNAs are able to regulate the factors involved in DNA methylation, as well as histone methylation/acetylation, imposing a broad range of control over the changing epigenetic landscape within aging MSC. Further works into elucidating the roles of different miRNA species and implicating them in not only aging but many different diseases will provide a greater insight into the ever-evolving epigenetic landscape.

Given the influence the environment has on our epigenome, future studies will increase dramatically, showing how diet and lifestyle affects the epigenetic drift and aging. Studies on high-fat/sugar diets have recently shown that a high-fat maternal diet can compromise the skeletal integrity of offspring, giving rise to low bone mass. MSC display senescence, reduced osteogenesis, increased adipogenesis and reduced glucose metabolism. Increased H3K27Ac on senescent genes and the repressive H3K27me3 on osteogenic genes, together with DNA methylation changes, drive senescence and inhibit osteogenesis. Further studies are needed to examine the plethora of histone and DNA modification changes in response to different diets and the effects of different metabolites that act as cofactors for epigenetic enzymes. One of the most effective treatments for bone loss is mechanical loading and resistance exercise, which has shown to result in reduced DNA methylation along osteogenic genes and the expression of long non-coding RNA species, enhancing osteogenesis. The information instructing the chromatin architecture and, therefore, cellular function and longevity is complex and interdependent. Just how all these layers change and contribute to aging represents a field that is now rapidly expanding with an effort to reboot the epigenetic code safely as a strategy to rejuvenate stem cells.

## Figures and Tables

**Figure 1 ijms-24-06499-f001:**
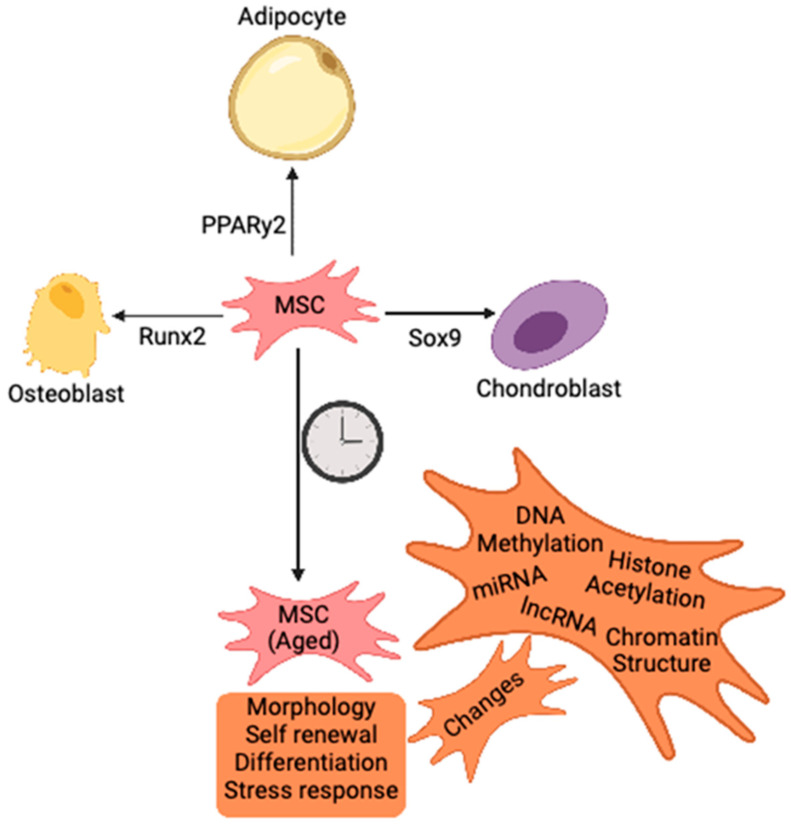
Molecular regulation of MSC cell fate. MSC differentiation is driven by a number of master regulators into the various lineages, including osteoblasts (Runx2), adipocytes (PPARy2) and chondroblasts (Sox9). Aged MSC show changes in phenotypes and differentiation properties, an effect of the changing epigenetic landscape within the cell.

**Figure 2 ijms-24-06499-f002:**
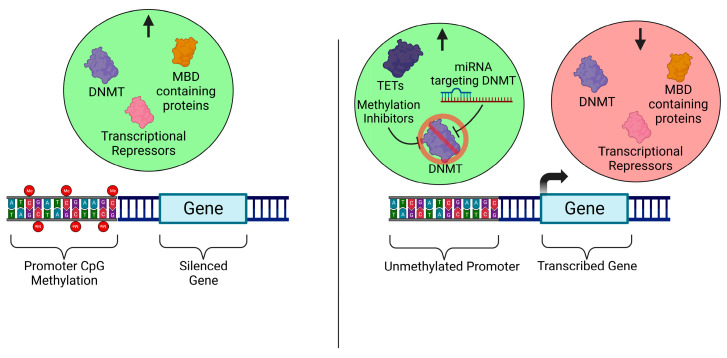
Regulators of DNA methylation. CpG methylation and the factors involved in writing (DNMT), reading (MBD containing proteins) and executing the silencing of genes (repressors). This mark is able to be removed or inhibited by the activity of DNA hydroxymethylases (TETs) and miRNA targeting DNMT. Receptor Activator of Nuclear Factor-kappa B (RANK).

## Data Availability

Not applicable.

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
