# Peer review of "Impact of Environmental and Epigenetic Changes on Mesenchymal Stem Cells during Aging"

_ijms, 2023, doi:10.3390/ijms24076499_

Round 1

Reviewer 1 Report

The authors of this article have described the epigenetic regulation of mesenchymal stem cells during aging process. The emphasis of the article is on the bone-related genes and processes with acknowledgement of effect of diets and mechanical loading on aging.

This is a well-written article with interesting details for readers. The author should consider the following points,

1)      There is a specific emphasis on bone within the article. However, cartilage with respect to stem cell aging and its DNA modification has not been described within this article, especially considering that the majority of bone tissues are derived from the process of endochondral ossificatio. Can the authors include more studies on cartilage and how this differs from bone with respect to stem cell aging ? This also needs to discussed throughout the article to contrast with bone.

2)      Mesenchymal stem cells are derived from different mesenchymal tissues (e.g. bone, synovium, adipose tissue). Are their differences in their DNA modifications in normal and aging situations ? If there are publications available then this needs to included.

3)      There is now a lot of research in the use of senolytic drugs for disease treatments. How can these be drugs be used to prevent mesenchymal stem cell aging in the DNA modification processes and thereby improve cellular therapies for musculoskeletal diseases ?

4)      Aging in mesenchymal stem cells is related to changes in cellular metabolism from a naturally glycolytic mechanism to oxidative phosphorylation that induces ROS generation. How do the cellular metabolism genes change with epigenetic processes in aging ? This has been discussed in the diet sections of the article, although this is an important stand alone section.

Author Response

22 March, 2023

Dr. Miomir Knezevic,

Guest Editor

International Journal of Molecular Sciences, Special Issue “Genomics and Epigenetics of Stem Cells”.

Re: Submission of Revised Manuscript ijms-1017002.R2

Dear Dr. Knezevic,

Please find enclosed our revised manuscript entitled “Impact of Environmental and Epigenetic changes on Mesenchymal Stem Cells during Aging” (ijms-2222659). We have addressed the reviewers’ concerns and comments below. Please find changes to the manuscript have been made in red text for R1 and highlighted in yellow for R2 responses.

Response to Reviewers Comments:

Reviewer #1

The authors of this article have described the epigenetic regulation of mesenchymal stem cells during aging process. The emphasis of the article is on the bone-related genes and processes with acknowledgement of effect of diets and mechanical loading on aging. This is a well-written article with interesting details for readers. The author should consider the following points,

(1) There is a specific emphasis on bone within the article. However, cartilage with respect to stem cell aging and its DNA modification has not been described within this article, especially considering that the majority of bone tissues are derived from the process of endochondral ossificatio. Can the authors include more studies on cartilage and how this differs from bone with respect to stem cell aging? This also needs to discussed throughout the article to contrast with bone.

Author Response: We appreciate the significance of cartilage and the effects of aging on chondrogenesis. The emphasis of this review is on bone marrow derived mesenchymal stem cells and osteogenesis. We feel the addition of chondrogenesis will greatly broaden the review to age related cartilage disorders and will significantly increase the word limit.

(2) Mesenchymal stem cells are derived from different mesenchymal tissues (e.g. bone, synovium, adipose tissue). Are their differences in their DNA modifications in normal and aging situations? If there are publications available then this needs to included.

Author response: We are focussing on bone marrow derived mesenchymal stem cells as they are the main contributors to bone formation. Including the other mesenchymal stem cells broaden the topic to other age related disorders other than bone and will be above the word limit. Nevertheless, we have added a new paragraph in Section 3.2 to indicate differences in methylation of key mesenchymal differentiation genes (Runx2, PPARg and Sox9) between MSC derived from adipose and bone marrow tissues.

(3) There is now a lot of research in the use of senolytic drugs for disease treatments. How can these be drugs be used to prevent mesenchymal stem cell aging in the DNA modification processes and thereby improve cellular therapies for musculoskeletal diseases?

Author response: Senolytics is a fascinating area, however it does not operate at the epigenetic level as far as mechanism is concerned. Its main course of action is inducing apoptosis in senescent cells. We have covered the epigenetic changes during senescence.

(4) Aging in mesenchymal stem cells is related to changes in cellular metabolism from a naturally glycolytic mechanism to oxidative phosphorylation that induces ROS generation. How do the cellular metabolism genes change with epigenetic processes in aging ? This has been discussed in the diet sections of the article, although this is an important stand alone section.

Author response: We agree with the reviewer on the ROS angle and have expand on this in section 4.2.

Reviewer # 2

Albeit the Authors wrote a reply letter stating the main differences between their previous published article on the same topic and the present one, the two reviews manuscripts are similar in several parts and the novelty of the present one is more detectable in what they have added in red color.

It is needed therefore to consistently reduce and delete all what it is not necessary in the present review,. also because well known to the scientific community, along with delete the majority of self citations (9!).

Same sections must be deleted or resumed or put together.

The Conclusions are inappropriate because they contain a Summary that is not needed: they provided a Graphical Abstract along with the abstract and both must be sufficient.

The Conclusions must be concise and must contain the "Perspectives"

The specific changes are specified to the Authors.

Author Response: We have removed the “summary” from the conclusions section. We appreciate the reviewer’s view but we maintain that this review is an updated version covering perspectives on epigenetic regulation of MSC with the addition of environmental and dietary factors. We feel that the number of self-citations is justified given the relevance of the publications in the field.

We are hopeful the Reviewer and Editorial Board can appreciate these differences and see the suitability of publishing this review.

Sincerely

Stan Gronthos

Reviewer 2 Report

This review summarises the epigenetic changes that occur during ageing and affect the differentiation capacity as well as the function of MSCs. The authors provide all the essential information in details explaining how DNA methylation patterns, histone modification changes and miRNAs differential expression have an impact on MSCs and how is this related to the development of the consequent diseases. On the other hand, they describe how environmental factors such as diet and exercise affect this procedure.

The manuscript is very well written and provides a comprehensive review of the current literature. In order to improve the quality of the manuscript, it would be suggested to include an explanatory illustration of which stage in the differentiation process of MSCs specific miRNAs affect the expression/function of transcription factors, like RunX2, Osx, Sox9, and direct the process towards a specific differentiation fate. In addition, it is also recommended to refer not only to HFD effects on epigenome of MSCs but also protein restricted diet based on recent publications. Otherwise, the manuscipt is sound and can be accepted for publication.

Author Response

Reviewer #1:

This review summarises the epigenetic changes that occur during ageing and affect the differentiation capacity as well as the function of MSCs. The authors provide all the essential information in details explaining how DNA methylation patterns, histone modification changes and miRNAs differential expression have an impact on MSCs and how is this related to the development of the consequent diseases. On the other hand, they describe how environmental factors such as diet and exercise affect this procedure.

The manuscript is very well written and provides a comprehensive review of the current literature. In order to improve the quality of the manuscript, it would be suggested to include an explanatory illustration of which stage in the differentiation process of MSCs specific miRNAs affect the expression/function of transcription factors, like RunX2, Osx, Sox9, and direct the process towards a specific differentiation fate. In addition, it is also recommended to refer not only to HFD effects on epigenome of MSCs but also protein restricted diet based on recent publications. Otherwise, the manuscript is sound and can be accepted for publication.

We thank the reviewer for their positive comments of our manuscript.

Qu 1.“In order to improve the quality of the manuscript, it would be suggested to include an explanatory illustration of which stage in the differentiation process of MSCs specific miRNAs affect the expression/function of transcription factors, like RunX2, Osx, Sox9, and direct the process towards a specific differentiation fate.”

Author Response:

We have now modified the text to provide the additional information concerning miRNA in red text in Section 3.5.

Qu 2. “In addition, it is also recommended to refer not only to HFD effects on epigenome of MSCs but also protein restricted diet based on recent publications.”

Author Response:

We have made the amendments to the review regarding protein restricted diet in red text in Section 4.1.

Reviewer 3 Report

The same Authors wrote and published a very similar Review Entitled:

"Epigenetic Regulators of Mesenchymal Stem/Stromal Cell Lineage

Determination" in Current Osteoporosis Reports (2020) 18:597605.

Do the Authors are capable to explain if the differences are great and the two papers are not very similar?

Author Response

Reviewer # 2

The same Authors wrote and published a very similar Review Entitled:

"Epigenetic Regulators of Mesenchymal Stem/Stromal Cell Lineage

Determination" in Current Osteoporosis Reports (2020) 18:597–605.

Do the Authors are capable to explain if the differences are great and the two papers are not very similar?

Author Response:

We thank the reviewer for their appraisal of the manuscript. We appreciate that we have written a review in the past about the role of epigenetics in stromal cell biology but this review is profoundly different to the one in question by the reviewer “Epigenetic Regulators of Mesenchymal Stem/Stromal Cell Lineage Determination" in Current Osteoporosis Reports (2020) 18:597–605”. Firstly the review published in 2020 is focussed on epigenetic determinants (DNA methylation/hydroxylation, histone modifications, non-coding RNA) of lineage specific differentiation of bone mesenchymal stem cells, whereas the present review is focused on the epigenetic changes (DNA methylation/hydroxylation, histone modifications, non-coding RNA) and determinants of mesenchymal stem cell aging and disease, hence it focuses on epigenetic changes during senescence and specifically age associated deregulation in self renewal and lineage determination. Secondly this review also incorporates the effect of diet on epigenetic changes and aging of mesenchymal stem cells and in bone disease. Thirdly the content on this review has focussed more so on the last 3 years covering more recent discoveries in the field.

We are hopeful the Reviewer and Editorial Board can appreciate these differences and see the suitability of publishing this review.

Round 2

Reviewer 1 Report

The authors have answered my review questions appropriately.

Reviewer 3 Report

Albeit the Authors wrote a reply letter stating the main differences between their previous published article on the same topic and the present one, the two reviews manuscripts are similar in several parts and the novelty of the present one is more detectable in what they have added in red color.

It is needed therefore to consistently reduce and delete all what it is not necessary in the present review,. also because well known to the scientific community, along with delete the majority of self citations (9!).

Same sections must be deleted or resumed or put together.

The Conclusions are inappropriate because they contain a Summary that is not needed: they provided a Graphical Abstract along with the abstract and both must be sufficient.

The Conclusions must be concise and must contain the "Perspectives"

The specific changes are specified to the Authors.

Author Response

(The authors gave the same response as above.)

Round 3

Reviewer 3 Report

Albeit the Authors answered to my previous concerns regarding the lack of novelty due to the previous already published manuscripts including one of the same Authors, this Review remains weak under all the aspects.

Therefore, this Academic Editor, serving as a reviewer in this occasion, is extremely doubtful on this review article.

At present, it is rather long and several sentences are redundant: a consistent length reduction is needed as well as a reduction of the number of cited articles that are excessive, including the self citations, whose number must not exceed the number of 3.